# Design of a Low-Frequency Dielectrophoresis-Based Arc Microfluidic Chip for Multigroup Cell Sorting

**DOI:** 10.3390/mi14081561

**Published:** 2023-08-05

**Authors:** Xueli Nan, Jiale Zhang, Xin Wang, Tongtong Kang, Xinxin Cao, Jinjin Hao, Qikun Jia, Bolin Qin, Shixuan Mei, Zhikuan Xu

**Affiliations:** 1School of Automation and Software Engineering, Shanxi University, Taiyuan 030006, China; 202123603033@email.sxu.edu.cn (J.Z.); 202123601010@email.sxu.edu.cn (X.W.); 202123603013@email.sxu.edu.cn (T.K.); 202222207001@email.sxu.edu.cn (X.C.); 202222207011@email.sxu.edu.cn (J.H.); 202222207015@email.sxu.edu.cn (Q.J.); 202222207026@email.sxu.edu.cn (B.Q.); 202223601010@email.sxu.edu.cn (S.M.); 202222207041@email.sxu.edu.cn (Z.X.); 2School of Biomedical Engineering, Shanghai Jiao Tong University, Shanghai 200030, China

**Keywords:** dielectrophoresis, cell sorting, curved channel, curved double side electrodes, cell movement trajectories

## Abstract

Dielectrophoresis technology is applied to microfluidic chips to achieve microscopic control of cells. Currently, microfluidic chips based on dielectrophoresis have certain limitations in terms of cell sorting species, in order to explore a microfluidic chip with excellent performance and high versatility. In this paper, we designed a microfluidic chip that can be used for continuous cell sorting, with the structural design of a curved channel and curved double side electrodes. CM factors were calculated for eight human healthy blood cells and cancerous cells using the software MyDEP, the simulation of various blood cells sorting and the simulation of the joule heat effect of the microfluidic chip were completed using the software COMSOL Multiphysics. The effect of voltage and inlet flow velocity on the simulation results was discussed using the control variables method. We found feasible parameters from simulation results under different voltages and inlet flow velocities, and the feasibility of the design was verified from multiple perspectives by measuring cell movement trajectories, cell recovery rate and separation purity. This paper provides a universal method for cell, particle and even protein sorting.

## 1. Introduction

With the development of the intersection of biomedicine and microchips, microfluidics is playing an increasingly important role in the field of biomedicine. Relying on the versatility of microfluidics in the microscopic field and the low consumables, easy processing, high efficiency and high stability of microfluidic chips, microchip laboratories provide a reliable environment for the study of microscopic substances such as cells and proteins. Microfluidic chips can perform many functions such as the screening of drug-resistant cells [1], detection of pathogens [2], focusing of particles or cells [3], mixing of particles [4], counting of fluorescent microspheres [5], immobilization of cells or protein macromolecules [6], protein detection [7], screening and isolation of DNA aptamers [8], etc. The cross-fertilization of microfluidics with other disciplines is the key to achieving many functions. As a fundamental part of biomedical research, cell sorting is also the key to important scientific fields such as pathology detection and antibody screening. The development of microfluidic chips has solved the problem of cell sorting by using expensive equipment and harsh experimental environments. Some of the most common principles to achieve cell sorting are filtration [9], optics [10,11], magnetic sorting [12] and dielectrophoresis [13] and acoustophoresis [14]; dielectrophoresis’ separation technology does not require the labeling of cells, in comparison to other methods, but also allows for rapid and continuous sorting and simpler chip fabrication processes and operation methods. In addition, dielectrophoresis enables the rapid and continuous separation of proteins, which is one of the reasons for the rapid development of dielectrophoresis. Different functions such as the concentration [15], capture [16], driving [17] and sorting [18] of cells can be achieved in microfluidic chips using the dielectrophoretic method. One of the necessary conditions for dielectrophoresis is the use of microelectrodes. The material and structure of the microelectrodes and their position on the chip affect the success rate of sorting. Depending on the microelectrode material, dielectrophoresis microfluidic chips are broadly classified into two types: electrode-based and insulator-based [3]. The insulator-based microfluidic chips generate a non-uniform electric field inside the channel by adding an insulator material to the structure inside the channel. The electrode-based dielectrophoresis microfluidic chips require an external electrode near the channel to generate a non-uniform electric field. The cells will be polarized by the non-uniform electric field to generate the dielectrophoretic force. Different structures of electrodes and channels are designed to perform different functions. Tianbo Gao et al. [19] proposed a novel design of a DC dielectrophoresis microfluidic chip with rectangular electrodes placed on one side of the channel horizontally and electrodes with rectangular teeth on the other side, and successfully separated Chlorella vulgaris coated with 100% PS particles at 3 μm and 6 μm. Pouya Sharbati et al. [20] proposed a novel microfluidic device that successfully isolated four types of cells, red blood cells, T-cells, U937-MC cells and Clostridium difficile bacteria, simultaneously by using two sets of electrodes, a cylindrical electrode and a sidewall rectangular electrode. Hien Vu-Dinh et al. [21] designed microfluidic chips with serpentine channels and arrays of counter electrodes and used cell-specific aptamers to enhance the capture intensity, greatly increasing the accuracy of capturing lung cancer cells in low concentration solutions. Yanjuan Wang et al. [22] designed four 3D rectangular stereoscopic electrodes, located at the four corners of the capture chamber, to achieve the capture and release of particles of specific diameters.

In studies of achieving the manipulation of a single class of cells or sorting of multiple cells, common electrodes can be divided into array electrodes [23], array pair electrodes [24,25] and interdigital electrodes [26], unilateral double electrodes [27] and bilateral asymmetric electrodes [28]. These electrodes require multiple sets of electrodes to be used simultaneously and are commonly used in through-channel type microfluidic chips as a way to meet the requirements of non-uniform electric fields. In addition to providing a non-uniform electric field in this way, there is also the method of pairing curved electrodes with curved channels to produce the desired electric field distribution. Machining curved electrodes on both sides of a curved channel allows for a simpler electric field distribution in the microchannel. This achieves the optimization of the dielectrophoretic force and effectively reduces material consumption when machining the electrodes.

Cells change their motion in the microfluidic chip due to forces and their forces are analyzed as gravity, buoyancy, dielectrophoretic force and fluid drag force. If the microchannel is designed to be horizontal and the electrodes are placed on its two sides, at which time the direction of dielectrophoretic force and the direction of fluid drag force are horizontal, and gravity and buoyancy are exactly in equilibrium in the vertical direction, the effects of gravity and buoyancy on the cell movement state can be greatly reduced. The principle of cell sorting based on dielectrophoresis is based on the difference of dielectric properties. This is also the key to the difference in dielectrophoretic force. Due to the different conductivity and dielectric constants of the cells themselves, the different conductivity and dielectric constant of the solution the cells are in, and the different size of the cells themselves, resulting in different dielectrophoretic forces on different cells. Then the fluid drag force provides the driving force towards the outlet direction of the microfluidic chip. Under the combined effect of these forces, the cells will move along the specified path and eventually discharge from the different outlet ports.

Completely designing a microfluidic chip requires all-round consideration. Its channel design basically determines the function of the microfluidic chip. And the design of the electrodes determines the different electric field distributions as well as the strength of the electric field. Choosing the processing materials and the buffers can also have a significant impact on the performance of the chip, and this impact may have the negative effect of limiting each other between different designs. Therefore, a number of issues must be considered when designing a microfluidic chip, such as the fact that the use of electrodes will cause the generation of joule heat inside the microfluidic chamber [29], leading to an increase in the temperature of the solution, which may destroy the activity of biological cells. In order to meet the requirements of the specified direction and size of the dielectrophoretic force, the distance between electrode pairs and the size of the microfluidic channel will be limited, leading to the limitation of the size of the microfluidic chip. A too short microfluidic channel also leads to insufficient cell movement. On the contrary, a too long microfluidic channel generates more joule heat to change the stability of the buoyancy force [30], and increases the instability of cell movement due to the unsmooth processing of the channel walls and the interaction between cells, resulting in the inability to reach the requirements of sorted cells.

To solve the above problems, some methods are proposed in this design. Firstly, an inhomogeneous electric field with uniform changes in potential gradient is created in order to solve the cell polarization phenomenon as much as possible. The array electrodes produce a repetitive electric field distribution by alternating positive and negative settings on one side, and their individual electrodes produce a fan-shaped uniform distribution of electric field contours, while a wave-like distribution of electric field contours can be produced by the array distribution. This potential distribution meets the requirements of dielectrophoresis, but causes the cells to pass through the left side of each electrode with dielectrophoretic forces in the opposite direction of advancement. This results in blocking the advance of the cells and is an extremely unstable and negative factor for cell sorting. Using 3/4 arc electrodes on both sides of the microchannel produces a more uniform potential gradient. And the resulting dielectrophoretic force does not cancel out the rest of the cell’s forward momentum. Combined with the 3/4 arc-shaped channel, the direction of the advancing force of the cells in the channel will constantly change, but always tangential to the dielectrophoretic force. It is necessary to maximize the role of the dielectrophoretic force in the process of cell deflection and maximize the effect of different dielectric properties of the cells. The cells are sorted by the negative dielectrophoretic force driving the cells to the outside of the channel. To make the cells have a more distinct differentiation path, this design adds a buffer inlet on the outside of the sample inlet (i.e., the outside of the circular channel) to pool the cell flow on the inside of the curved channel. Based on the laminar flow in the microchannel, cells will move slowly along the inner wall of the channel. With the addition of dielectrophoretic force, cells with different dielectric properties will be deflected to different degrees.

In this paper, a dielectrophoretic microfluidic chip with smaller channel size and more optimized electric field distribution was successfully simulated by coupling the effects of chip size, channel shape, voltage amplitude and fluid flow rate on cell sorting. It could quickly complete the sorting of multiple groups of cells. This paper will analyze the effects of various factors on the feasibility and sorting effect of microfluidic chips, providing a complete set of simulation flow. The reader can review relevant information beyond the data provided in this paper, and then follow the simulation flow of this paper to perform cell sorting experiments. Theoretically, if the particle model of the protein can be successfully calculated, the design will be able to tune the parameters such as voltage amplitude, frequency or flow rate to complete the protein sorting. Protein sorting will be more effective if the protein is amplified by polymerase chain reaction or isothermal amplification techniques before being fed into the microfluidic chip.

## 2. Theoretical Concept

The premise of the dielectrophoretic force is that the particle or cell is polarized in a non-uniform electric field. The phenomenon of cell polarization is the accumulation of charges inside the cell to produce induced charges on the cell surface which, in turn, changes the dipole moment of the induced charge inside the cell. The presence of a dipole moment interacts with the electric field to produce dielectric swimming forces which, in turn, change the motion of the cell. As shown in Figure 1a, the particles P1 and P2 in the figure are two types of cells possessing different polarization rates, and the different polarization rates lead to different degrees of polarization. The polarized cells generate dielectrophoretic forces in a non-uniform electric field, which can be understood as the difference in direction and magnitude of dielectrophoretic forces due to the difference in dielectric properties of different cells. Particle P1 is subjected to negative dielectrophoretic force, and the direction of dielectrophoretic force is toward the low electric field strength. Due to the advantage of 3/4 arc electrode structure, the direction of electric field strength is the same as the direction of electric field line, which can be regarded as the direction of dielectrophoretic force points to the negative electrode along the direction of electric field line. Particle P2 is subjected to positive dielectrophoretic force and the direction of force is opposite to particle P1.

The equation for the dielectrophoretic force of a particle in an alternating current is:(1)$$F_{dep}=2\pi \varepsilon_{m}r^{3}Re\left[K_{cm}\right]\nabla {\left|E\right|}^{2}$$
where $\varepsilon_{m}$ denotes the dielectric constant of the solution, r denotes the particle radius, $\left|E\right|$ denotes the root mean square of the AC electric field strength and $Re\left[K_{cm}\right]$ denotes the real part of the Clausius–Mossotti (CM) factor:(2)$$Re[K_{cm}]=Re\left[\frac{\varepsilon_{p}^{*}-\varepsilon_{m}^{*}}{\varepsilon_{p}^{*}+2\varepsilon_{m}^{*}}\right]$$

The complex dielectric constant of the particle is:(3)$$\varepsilon_{p}^{*}=\varepsilon_{p}-j\sigma_{p}/\omega$$

The complex dielectric constant of the solution is:(4)$$\varepsilon_{m}^{*}=\varepsilon_{m}-j\sigma_{m}/\omega$$
where $\varepsilon_{p},\varepsilon_{m}$ are the dielectric constant of the particle and the solution, respectively, $\sigma_{p}{,\sigma }_{m}$ are the conductivity of the particle and the solution, respectively, and ω is the current frequency.

Bringing Equations (3) and (4) into Equation (2), this can be reduced to:(5)$$Re[K_{cm}]=\frac{\left(\varepsilon_{p}-\varepsilon_{m}\right)\left(\varepsilon_{p}+2\varepsilon_{m}\right)+\frac{\left(\sigma_{p}-\sigma_{m}\right)\left(\sigma_{p}+2\sigma_{m}\right)}{\omega^{2}}}{{\left(\varepsilon_{p}+2\varepsilon_{m}\right)}^{2}+\frac{{\left(\sigma_{p}+2\sigma_{m}\right)}^{2}}{\omega^{2}}}$$

According to Equation (5), it can be found that:(6)$$\underset{\omega \to 0}{\lim}Re[K_{cm}]=\frac{\sigma_{p}-\sigma_{m}}{\sigma_{p}+2\sigma_{m}}$$
(7)$$\underset{\omega \to \infty }{\lim}Re[K_{cm}]=\frac{\varepsilon_{p}-\varepsilon_{m}}{\varepsilon_{p}+2\varepsilon_{m}}$$

Therefore, when the frequency tends to 0, the conductivity determines the cell CM factor, and when the frequency is very high the dielectric constant is the decisive factor. When $Re[K_{cm}]>0$, the positive dielectrophoretic force drives the particle to the position with the maximum electric field strength, and when $Re\left[K_{cm}\right]<0$, the negative dielectrophoretic force drives the particle to the position with the minimum electric field strength. The CM factor can be calculated using Equation (2) for particles of the membrane-free spherical model such as polystyrene microspheres. Cells differ from such particles in that they have a cell membrane, which is a monolayer membrane structure composed of phospholipid bilayers, the cell membrane components differ greatly from those of the cytoplasm, so the influence of the cell membrane cannot be ignored. And the conductivity and dielectric constant of the cell membrane are different from those of the cytoplasm, so using Equation (2) only to calculate the CM factor of cells will cause a large deviation. After numerical simulation, if the CM factor is calculated by using the cell as a model without membrane spheres, the value of $Re[K_{cm}]$ will change from negative to positive, at which time the cell will receive a positive dielectrophoretic force, contrary to the actual physical phenomenon. In order to restore the dielectrophoresis of the cell in a solution to a higher degree, the cell can be equated to a single-shell model [14], which is shown in Figure 1b.

The following equation is usually used instead of $Re[K_{cm}]$ of the particle:(8)$$Re\left[K_{cm}\right]=Re\left[\frac{\varepsilon_{c}^{*}-\varepsilon_{m}^{*}}{\varepsilon_{c}^{*}+2\varepsilon_{m}^{*}}\right]$$
where $\varepsilon_{c}^{*}$ is the single-shell model complex dielectric constant of the cell:(9)$$\varepsilon_{c}^{*}=\varepsilon_{2}^{*}\frac{{\left(\frac{r_{2}}{r_{1}}\right)}^{3}+\frac{2\left(\varepsilon_{1}^{*}-\varepsilon_{2}^{*}\right)}{\varepsilon_{1}^{*}+2\varepsilon_{2}^{*}}}{{\left(\frac{r_{2}}{r_{1}}\right)}^{3}-\frac{2\left(\varepsilon_{1}^{*}-\varepsilon_{2}^{*}\right)}{\varepsilon_{1}^{*}+2\varepsilon_{2}^{*}}}$$
where $r_{1}$ and $r_{2}$ are the radius of the cell without membrane and the radius of the cell with cell membrane, respectively. And ${\tilde{\varepsilon }}_{1}$ and ${\tilde{\varepsilon }}_{2}$ are the complex dielectric constant of the cell and the complex dielectric constant of the cell membrane, respectively. The complex dielectric constant can be calculated using Equation (3), and the cell conductivity, cell dielectric constant, cell membrane conductivity, and cell membrane dielectric constant are referred to as $\sigma_{1}$, $\sigma_{2}$, $\varepsilon_{1}$ and $\varepsilon_{2}$. The dielectrophoretic force of the single-shell model can be calculated by directly applying Equations (8) and (9) to Equation (1).

The cells flow in from the entrance and, under the action of the buffer, the cells move close to the inner wall of the channel. The role of the dielectrophoretic force is to provide power to the cells toward the outer wall of the channel. And it distinguishes the cells into several cell streams, each cell stream containing only one kind of cells, so the dielectrophoretic force only changes the offset of the cells, from beginning to end. When the cells are in a fluid, the fluid drag force provides the forward power for them. Because the fluid Reynolds number is small, the fluid is in the laminar flow state, so the direction of the fluid drag force is also more stable. The fluid drag force equation is:(10)$$F_{drag}=\frac{1}{\tau_{p}}m_{p}\left(u-v\right)$$
(11)$$\tau_{p}=\frac{\rho_{p}d_{p}^{2}}{18\mu }$$
where $\tau_{p}$ is the cell relaxation time, $m_{p}$ is the mass of the cell, u and v are the velocity of the fluid at the cell location and the velocity of the cell, respectively, $\rho_{p}$ is the cell density, $d_{p}$ is the cell diameter and μ is the dynamic viscosity of the fluid.

In addition to the dielectrophoretic force and fluid drag force, there are buoyancy, gravity and Brownian force in the solution. The fluid flow direction in the microfluidic channel designed in this paper is horizontal, so the influence of buoyancy and gravity on the cell sorting effect is very small. In addition, the density of the buffer is similar to that of the cells; the gravity is balanced with the buoyancy, so as to reduce the influence of gravity and buoyancy on the cells in the vertical direction and prevent the cells from colliding with the upper and lower walls. The Brownian force is generated due to the irregular motion of molecules hitting the cell. The magnitude of the Brownian force is related to the size of the force object and the temperature of the environment, the smaller the particle size and the higher the ambient temperature, the larger the Brownian force, whose effect is negligible in this paper. So the force on the cells can be analyzed as the combined force of dielectrophoretic force and fluid drag force, and the force analysis is shown in Figure 1c: in the x-y plane, as the cell migrates in the position of the channel, the direction of the fluid drag force is constantly changing with the direction of fluid advancement, the direction of dielectrophoretic force is always the same as the direction of electric field lines, and the strength of dielectrophoretic force also changes. In the figure, *F*_1_, *F*_2_, *F*_3_ are the combined forces of fluid drag force and dielectrophoretic force. In the y-z plane, the buoyancy force $F_{bouy}$ and gravity G of cells at different heights are always in balance, so that similar cells at all horizontal heights are subject to the same dielectrophoretic force. And the cells rely on fluid drag force to provide power to flow to the channel outlet from the beginning to the end.

## 3. Design and Simulation

### 3.1. Chip Structure Design

The microfluidic chip designed in this paper is a circular channel, as shown in Figure 2a, with two inlets to inject the slower flowing sample solution and the faster flowing buffer solution, and three outlets are provided to sort out two or three types of cells. As shown in Figure 2b, the electrodes of circular shape are set on both walls of the channel, which can form a simpler and more stable electric field gradient compared with the common array electrodes and interdigital electrodes. From Equation (1), it is known that the electric field gradient is one of the important factors to determine the magnitude of the force. When the cells flow with the buffer, the rate of change in the electric field gradient at which they are located is always unique, which can form a more stable dielectrophoretic force for the cells.

The dimensions of the channels in Figure 2a are listed in Table 1. In the table, L1-7 denotes the width of the microchannel, R1, R2 denote the radius of the circle and θ1-4 denote the angle of the microchannel outlet. Where θ1 is fixed to 90°, this indicates that the arc of the microfluidic chip is a 3/4 circular arc. θ2-4 indicate the angle of inclination of the side outlet, and the sum of θ2 and θ3 is 90°. Where the larger θ3 and θ4 are, the faster the fluid changes direction when passing through the outlet position, which may cause the cells to collide on the channel wall under the effect of inertia, and the smaller they are, the higher the processing difficulty will be. So the value of θ2-4 can be taken as 45°.

Currently, there are many methods regarding the processing and fabrication of microfluidic chips containing metal electrodes. Yanjuan Wang et al. [22] used glass to fabricate the substrate, polydimethylsiloxane(PDMS) to fabricate the microfluidic channel, Ag-PDMS to fabricate the metal electrodes, indium-tin-oxide (ITO) to fabricate the wires and processed the chip using soft lithography. Misaki Hata et al. [17] processed ITO electrodes using conventional lithography using a double adhesive film of 30 μm in thickness to fabricate the fluidic channel. Avijit Barik et al. [16] accomplished the processing of graphene electrodes through a multiple process by first fabricating a metal electrode (Ti/Pd) on a silicon plate with a pattern on a silicon plate, then a layer of HfO_2_ was deposited using atomic layer deposition (ALD), and a single layer of graphene was grown by chemical vapor deposition (CVD). Finally, photolithography was employed to pattern Cr/Au contact electrodes. Soft lithography can process the electrode structure in micrometer scale and chemical vapor deposition (CVD) can process the electrode structure in nanometer scale. According to Table 1, it can be seen that the microchannel of the microfluidic chip is 30 μm at the narrowest point, which is the achievable processing size. In this paper, the simulation experiments use the common PDMS as the material of the microfluidic channel. Considering that Cu has superior plasticity and excellent stability when making metal electrodes, Cu is used as the metal electrode material for simulation.

### 3.2. CM Factor Simulation

From the dielectrophoretic force equation, it can be seen that the real part of the CM factor determines the direction of the dielectrophoretic force, and the CM factor is determined by the combination of conductivity, dielectric constant and electric field frequency, etc. The CM factor was calculated using MyDEP [31], and Figure 3a–c show the comparative CM factor values of various cells such as platelets, erythrocytes [32], leukocytes and circulating tumor cells [33] in buffers with conductivities of 0.005 S/m, 0.055 S/m and 0.5 S/m, respectively. The dashed line in the figure is the zero datum line, marking the change in the direction of the dielectrophoretic force at that value. The microfluidic structure pools the cells in the inner side of the curved channel and uses the negative dielectrophoretic force to drive the cells toward the outer side of the channel to the designated position. So the small conductivity, as shown in Figure 3a, cannot provide negative dielectrophoretic force for all cells. As shown in Figure 3c, high conductivity can provide stable negative dielectrophoretic force for all cells, but high conductivity generates more joule heat. The thermal effect can disrupt the structure of laminar flow and affect the direction of fluid drag force in the channel. And fluid temperatures above 42° also destroy the activity of the vast majority of human cells.

The buffer used for the simulation has a dielectric constant of 80. Table 2 counts the eight cells included in Figure 3, namely platelets, erythrocytes, sphere-like leukocytes, A549 circulating tumor cells, MCF-7 non-metastatic breast cancer cells, MCF-10A non-tumor breast epithelial cells, HT-29 human colon cancer cells and MDA-MB-231 human breast cancer cells. The information listed is cell radius $r_{2}$, cell conductivity $\sigma_{1}$, cell dielectric constant $\varepsilon_{1}$, cell membrane conductivity $\sigma_{2}$, cell membrane dielectric constant $\varepsilon_{2}$ and source of data. The eight cells counted in the table will be used in Section 4.2 to validate the performance of the microfluidic chip in processing different groups of mixed cell solutions. Erythrocytes, leukocytes and platelets are used as simulations for the isolation of common blood cells. A549 and three blood cells are used to experiment the isolation effect at different diameters. MCF-7, MCF -10A and HT-29, MDA-MB-231 are used to experiment the isolation effect when the diameters are similar but the dielectric properties are different. In addition, HT-29, a rare cancer cell, was isolated from blood.

### 3.3. Electric Field Simulation

According to the dielectrophoretic force formula, it is known that the gradient change in the electric field affects the magnitude of the dielectrophoretic force. The chip is designed with curved electrodes set on both walls of the channel. To study the effect of the changing electric field gradient situation on cell sorting, electric field simulation using COMSOL Multiphysics can produce the potential distribution shown in Figure 4a. The effect of voltage on the sorting performance of microfluidic chips of specific sizes was investigated by simulating the trajectory tracking of particles at voltages of 10 V, 15 V and 20 V. The experimental objects contain three types of cells, platelets, erythrocytes and spherical leukocytes, and the simulation results are shown in Figure 4b–d. In Figure 4b, since the voltage of 10 V at a channel width of 100 μm is not enough to generate proper dielectrophoretic force, resulting in the cells not being able to move to the proper position in the short channel in time. Spherical leukocytes with larger particle size will be discharged from the middle outlet, while platelets and erythrocytes with smaller particle size will be discharged from the right outlet. As in Figure 4c, the voltage amplitude was first increased to 15 V, the cells would be differentiated significantly with the increase in voltage leading to the motion trajectory. At this point, spherical leukocytes would be successfully discharged at the left outlet, but the distinction between the other two small diameter cells was still not obvious enough, and erythrocytes could be observed at the middle and side outlets. As shown in Figure 4d, the voltage was increased to 20 V, and the cell sorting effect was very obvious. Three types of cells would be discharged from different outlets, but the larger size leukocytes would appear to collide with the outer side of the channel. Since the inner wall of the actual microfluidic channel cannot be processed to be absolutely smooth, the collision effect between the channel wall and the cells will greatly affect the cell sorting results. Finally, 19 V was chosen as the optimal solution to drive the cells to the established trajectory while avoiding collision with the inner wall of the channel as much as possible in the process of cell driving. The distribution of cells in the channel is shown in Figure 4e. In order to more clearly describe the trajectory of cells in the channel, the paths of cells are represented by lines as shown in Figure 4f. It can be observed that the trajectories of different kinds of cells in the arc microfluidic chip are gradually differentiated and finally completely separated.

### 3.4. Flow Rate Simulation

The microfluidic chip set up in this thesis uses two inlets, buffer and sample solution are injected from channel inlet 1 and inlet 2, respectively, and the two or three cell-sorted solutions are collected from the three outlets. The flow rate difference between the buffer and the sample solution determines whether the cells will be collected on the inside or outside of the channel. If the buffer flow rate is higher, the cells will be collected on the inside of the channel, and vice versa if the sample flow rate is higher. In this paper, we chose a low frequency band; several cells were subjected to negative dielectrophoretic force, which drove the cells to move to the side with weaker field strength. So the flow rate of buffer should be larger than the flow rate of sample solution, and the larger the flow rate difference, the closer the cells are to the inner channel wall. The flow velocity simulation using COMSOL Multiphysics, Figure 5 shows the flow velocity in the channel when the flow velocities of inlet 1 and 2 are 0.56 mm/s and 2.4 mm/s, respectively. The simulation uses a small Reynolds number of the solution, and the fluids in the channel are all laminar, without the generation of turbulence, which can provide a stable fluid drag force for the cell movement.

Since the fluid in the channel is laminar, the difference in flow velocity between inlet 1 and inlet 2 pools the cells in the sample fluid on the inner side of the channel and provides a stable direction of fluid drag force. As shown in Figure 6, where the two inlets of each microfluidic chip intersect, the cells are subjected to negative dielectrophoresis towards the outside of the channel, thus differentiating different trajectories of movement. To investigate the effect of the difference in flow velocity between the two inlets on the cell sorting results in a given size of microfluidic chip, this paper sets up simulations to increase the flow velocity of inlet 1 from 1.5 mm/s to 7 mm/s and the flow velocity of inlet 2 from 0.37 mm/s to 0.75 mm/s to observe the effect on the performance of the microfluidic chip. Figure 6 shows the results of nine sets of simulations with inlet 1 flow rate of 1.5 mm/s, 3 mm/s, 7 mm/s and inlet 2 flow rate of 0.37 mm/s, 0.56 mm/s, 0.75 mm/s. The cells used for the flow rate simulation experiments were also platelets, erythrocytes and spherical leukocytes. As shown in Figure 6a–i, 1.5 mm/s vs. 0.37 mm/s, 0.56 mm/s, 0.75 mm/s; 3 mm/s vs. 0.37 mm/s, 0.56 mm/s, 0.75 mm/s; 7 mm/s vs. 0.37 mm/s, 0.56 mm/s, 0.75 mm/s, respectively. When the inlet 1 flow velocity is small, as shown in Figure 6a–c, regardless of the variation of inlet 2 flow velocity, this causes all cells to move too far to the outside of the channel and deviate from their desired position. The reason for this situation is that the cells are subjected to negative dielectrophoretic forces for a longer period of time inside the channel at low flow velocities. On the contrary, as shown in Figure 6g–i, if the inlet 1 flow velocity is too large it does not provide enough time for the cells to move and results in the discharge of many kinds of cells from the inner outlet port. As shown in Figure 6d–f, the microfluidic chip can achieve the sorting purpose when the inlet 1 flow rate is 2.4 mm/s. In order to make the sorting rate faster, the inlet 2 flow rate can be increased without affecting the sorting effect. It can be concluded that, when the difference between the inlet 1 and inlet 2 flow rates is large, the inlet 1 flow rate can be used to directly control the trajectory of the cells, and the inlet 2 flow rate can be used to control the sorting rate of the cells. Its specific feasible maximum rate still needs experimental verification. The ideal flow rates of 2.4 mm/s and 0.56 mm/s were finally selected to provide a more stable flow rate for cell sorting.

### 3.5. Joule Heat Simulation

The microfluidic chip based on dielectrophoresis uses the solution in the channel as a conductive medium, and after the electrodes are energized, the microfluidic chip generates joule heat. The continuous use of the chip will cause the solution to warm up and even cause deformation of the microfluidic chip. The equation of joule thermal power is Q=J·E, J is the electric fluid density vector and E is the electric field strength vector. So the increase in voltage amplitude and frequency will generate more joule heat to some extent. Usually cells can survive for a certain time at a temperature no higher than 315.15 K. To avoid the situation that the solution temperature is too high to destroy the cell activity, this design has the following advantages to reduce joule heat generation:The buffer conductivity is 55 mS/m, and low conductivity is the most fundamental measure to reduce joule heat generation;The use of small amplitude low frequency AC at 19 V and 40 kHz;The minimum channel length (inner sidewall length) is 472 um, the maximum length (outer sidewall length) is 942 um and the two inlet flow rates are 0.56 mm/s and 2.4 mm/s, so that the cell stays in the channel for only 0.26~0.29 s.

Joule heat simulation was performed using COMSOL Multiphysics, and the experiments were set up with the channel walls as thermal insulation and the solution thermal conductivity coefficient was 0.5 W/(m·K) and the constant pressure heat capacity was 4.2 kJ/KG·°C. All the heat generated by the joule thermal effect will be collected inside the microfluidic chip to avoid heat dissipation from both sides of the channel wall, so as to more accurately evaluate the heat change in the solution inside the microfluidic channel. Since the fluid inside the channel is in a flowing state, the joule heat will be linearly related to the length of time the fluid is used as a conducting medium. The results are shown in Figure 7, where the entrance shows the lowest temperature, and the temperature increases as the fluid advances in the direction. The temperature can reach up to 344.7 K. The simulation experiment is calculated under the condition of adiabatic channel wall. The actual test can provide good heat dissipation conditions for the microfluidic chip due to the thermal conductivity of the processed material.

### 3.6. Different Size Simulation

The design of the curved channel with curved electrodes becomes a feasible solution through the analysis of voltage, flow rate and joule heat. In order to investigate the optimal size of the channel, this paper designs experiments to perform comprehensive simulations of microfluidic channels with different inner radii and different widths. From the size of the microfluidic chip, the electric field amplitude, the flow rate and the joule heat at each size are investigated to accomplish the sorting task. The experiments were set up with a minimum inner diameter of 100 μm and a maximum inner diameter of 400 μm, and 11 sets of simulation experiments with different channel widths were set up every 10 μm, increasing the channel width from 100 μm to 200 μm. The size simulation experiments still use platelet, erythrocyte and spherical leukocyte models as experimental subjects. The performance data include voltage, maximum temperature variation and average sorting time, all of which are the optimal parameters feasible in a single simulation. Among them, the temperature change in the simulation experiment is the maximum temperature change at zero bulk heat, but the effect of fluid flow as well as fluid heat transfer on the temperature change has been calculated. So it can be fully used as a reference for the performance of the microfluidic chip. The average sorting time is set to the initial position of the cells at the entrance at random, and the initial velocity of each cell is calculated by the flow field velocity field calculation method described in the previous section. One cell is released every 0.01 s from 0 s to 1 s. A total of 100 cells of each type are released, and the average time of their presence in the channel is calculated. As shown in Figure 8, it can be observed that the increase in the inner channel diameter or channel width is accompanied by a different increase in voltage, temperature and average time. The results indicate that increasing the channel width has a more significant impact on the overall performance than increasing both the inner and outer radius.

## 4. Results

### 4.1. Feasibility Verification

In order to visualize the sorting of the microfluidic chip, particle tracking statistics were added to the simulation. The wall condition was set to freeze before the simulation to exclude the cells that collide with the channel wall and may be damaged. The particle release duration at the entrance was set to 1 s, the time interval was set to 0.01 s, the initial position was set to random, and the number of any kind of blood cells was 100, which could fully simulate the random state of blood cells at the entrance of the channel in the actual test. As shown in Figure 9a, the particles started to appear at the outlet after 0.26 s, and all three blood cells reached the outlet by 1.27 s. Cell recovery and outlet purity are shown in Figure 9b, with 101%, 99% and 100% cell recovery for the three outlets, respectively. The total number of cells released agrees with the cell recovery overview indicating that the cells did not collide with the channel wall. However, the reason for the recovery not being all 100% is that one of the 100 released erythrocytes appeared at the location of outlet 1. The reason for this is that the erythrocyte trajectory is a line with a width, and very few erythrocytes located on both sides of the trajectory will flow out of the other outlets. Therefore, in the simulation experiment, we should try to ensure that the cell movement trajectory passes through the middle position of the outlet. The cell purity at the outlet of the microfluidic chip was 99% at outlet 1, 100% at outlet 2, and 100% at outlet 3, which proved the feasibility of the chip for efficient sorting.

### 4.2. Selectivity Verification

According to Figure 3b, it can be seen that, at a solution conductivity of 0.055 S/m, almost all cells were affected by negative dielectrophoresis in the low frequency band. It can be guessed that these cells can be sorted using the microfluidic chip designed in this paper. Four sets of simulation experiments were completed by COMSOL Multiphysics: the separation of circulating tumor cells in blood, sorting of MCF-7 and MCF-10A, separation of human HT-29 (without red blood cells) in blood and sorting of HT-29 and MDA-231. The number of cell types included in the experiments ranged from 2 to 4. The specific parameters and results configured in the simulation are listed in Table 3. The general parameters of the simulation experiment were: electric field frequency of 40 kHz, buffer dielectric constant of 80 and inlet 2 flow rate of 0. 56 mm/s. In the simulation of separation of circulating tumor cells in blood, outlet 1 discharged cells with 23% erythrocytes and 77% platelets. Outlet 2 cells were 43% erythrocytes and 57% leukocytes. Outlet 3 cells were 100% A549. The main reason for the mixing of outlet 1 and outlet 2 cells is that the microfluidic chip is three outlets, and the trajectory of erythrocytes is located between leukocytes and platelets, resulting in erythrocytes being discharged from both outlets. In the analysis simulation of MCF-7 and MCF-10A, no cells were discharged from outlet 1, 40% of MCF-7 and 60% of MCF-10A in outlet 2 and 60% of MCF-7 and 40% of MCF-10A in outlet 3, which did not achieve the separation effect. In the simulation of isolation of colon cancer cells HT-29, outlet 1 contained 50% platelets and 50% T-cells, outlet 2 had no cells discharged, and outlet 3 HT-29 contained 100%. This is because HT-29 cells undergo positive dielectrophoresis at a buffer conductivity of 0.055 S/m. In order to change the direction of the HT-29 dielectrophoretic force, it was necessary to increase the buffer conductivity to 0.1 S/m. At this point, the dielectrophoretic force exerted on platelets and T cells became weaker resulting from the change in solution conductivity. In the simulation of the separation of HT-29 from MDA-231, both outlet 2 and outlet 3 contained a large number of both types of cells, and the separation effect was not accomplished.

The curved dielectrophoresis microfluidic chip designed in this paper was verified to work for some cell mixtures through multiple sets of simulation experiments on the microfluidic chip. Due to the channel length, this design was not able to provide enough movement time for cells when separating cells with close diameters. It is possible to incorporate cell specificity to cause a change in the diameter or mesopoint characteristics of the cells. Once clearly differentiated from other cells, they can be applied to the arc dielectrophoresis microfluidic chip. When the chip handles more than three cell separations, there is a problem of mixed cell expulsion, and experiments cannot be performed when the dielectric properties of the cells are unknown. A mini-microscope can be integrated for observation, and the experiment can be artificially regulated by combining inlet flow rate control and voltage control. This method does not require the measurement of the dielectric properties of the cells, but requires extensive experimentation to find the proper flow rate and voltage. In the future, it is also possible to use convolutional neural networks to repeatedly learn the movement of different cells under different dielectrophoresis forces, and analyze the characteristics of the cells in real time by combining with image processing, to realize the intelligent control of the flow rate and voltage, and ultimately to integrate into an integrated intelligent cell sorting system.

## 5. Discussion

In this paper, a novel low-frequency microfluidic structure based on dielectrophoresis was demonstrated, and the effects of different structural parameters on the performance of microfluidic chips were investigated. The optimal parameters were selected by numerical simulation, which not only optimized the cell separation rate and efficiency from the structure, but also greatly reduced the effect of joule heat on cell activity. The damage caused by the repeated collisions of cells within the channel wall is avoided. Therefore the structure and method can effectively improve the cell survival rate. The microfluidic chip has a compact structure, which makes it easier to integrate with other devices. In addition, the throughput of the microfluidic chip can be increased by increasing the depth of the channel to meet the optimization in speed and efficiency. Due to the special characteristics of the microfluidic chip designed in this paper, its metal electrodes are located on both sides of the microfluidic channel. And the main effect of its generated electric field is reflected in the change in electric field gradient in the horizontal direction. By increasing the height of the microchannel and the electrodes at the same time, the distribution of the electric field gradient and the dielectrophoretic force applied to the cells will not change theoretically. There is some promise for increasing the speed and efficiency of microfluidic chips. This paper can provide some opinions for biomedical research. Since the study only used COMSOL Multiphysics software for simulation experiments, it failed to use real biological samples for experiments. Possible errors in channel smoothness, cell activity and fluid movement can affect the performance of the chip, and further experiments are needed to verify the reliability of the design.

## Figures and Tables

**Figure 1 micromachines-14-01561-f001:**
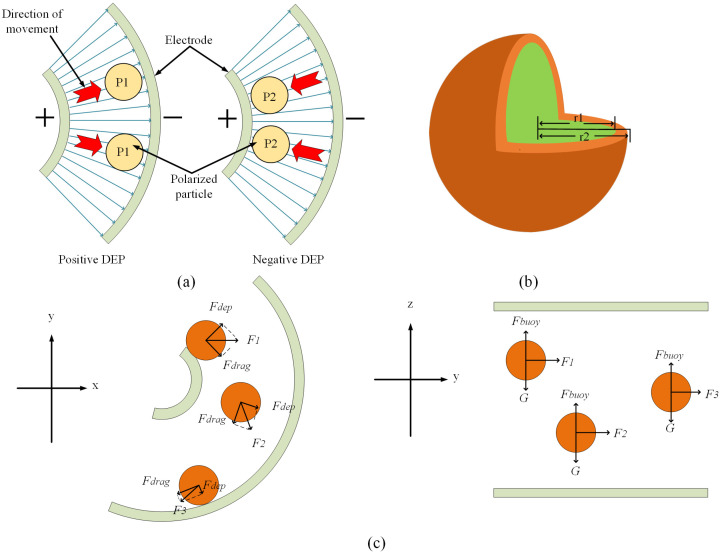
(**a**) Polarization phenomenon of particles in a non-uniform electric field; (**b**) Single-shell model of a cell; (**c**) Force analysis of a cell in a channel in the horizontal and vertical directions.

**Figure 2 micromachines-14-01561-f002:**
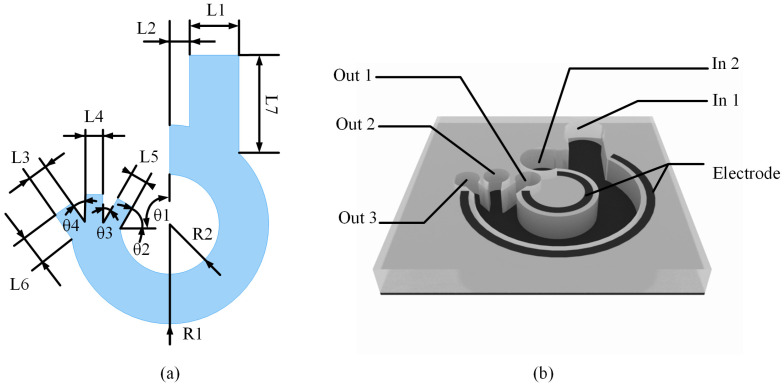
(**a**) Microfluidic chip channel size labeling; (**b**) 3D schematic of the microfluidic chip containing electrodes.

**Figure 3 micromachines-14-01561-f003:**
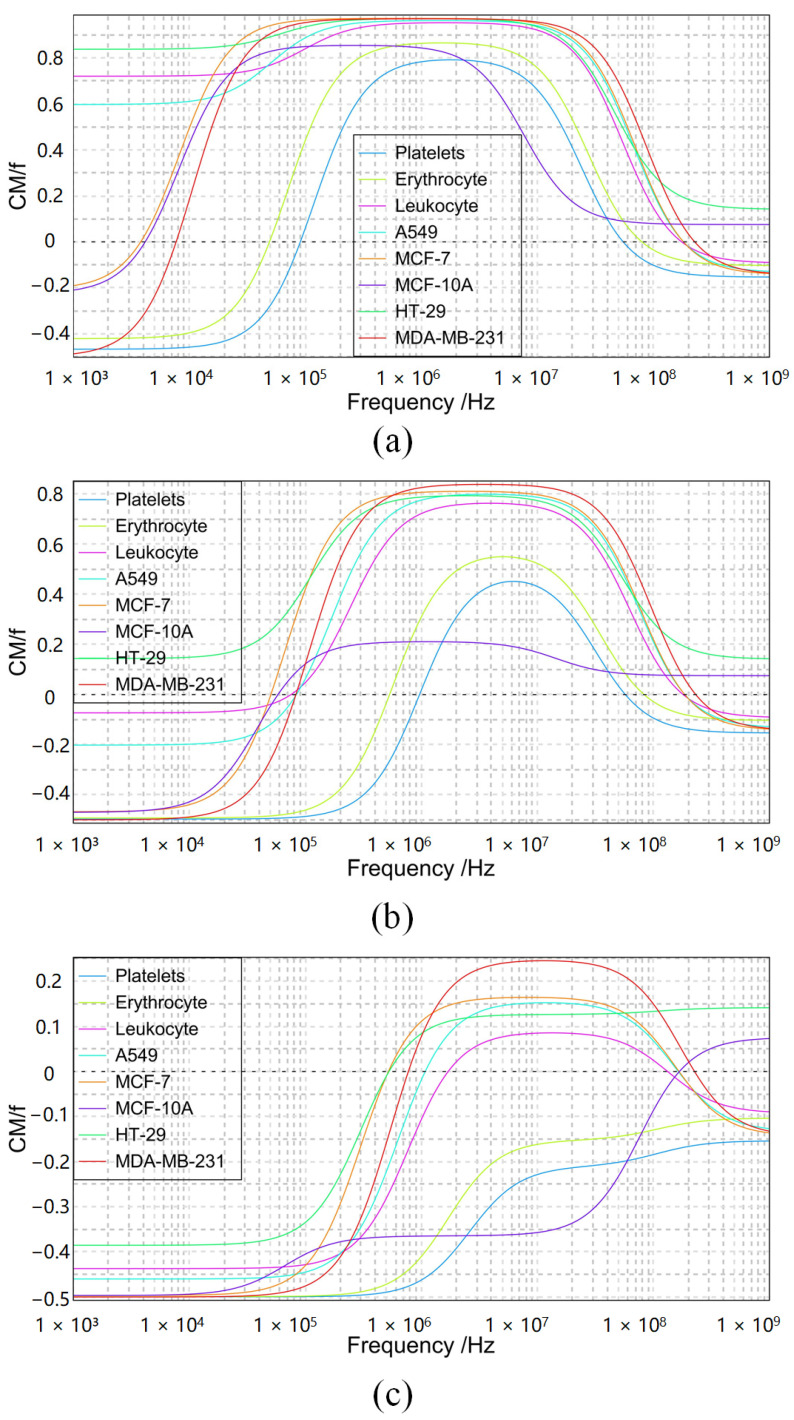
Simulation of CM factors of eight cells under different solution conductivities, (**a**) 0.005 S/m; (**b**) 0.055 S/m; (**c**) 0.5 S/m.

**Figure 4 micromachines-14-01561-f004:**
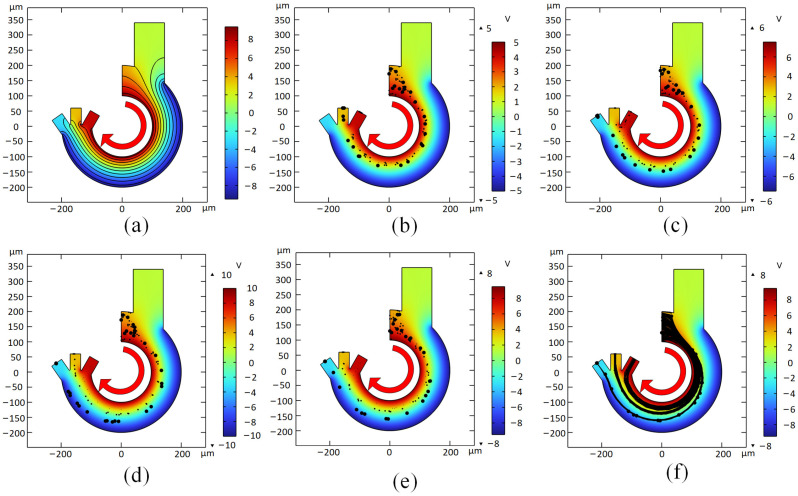
(**a**) Electric field distribution; (**b**–**e**) Cell sorting at 10 V, 15 V, 20 V and 19 V; (**f**) Cell movement trajectory at 19 V. Arrows indicate the direction of liquid flow.

**Figure 5 micromachines-14-01561-f005:**
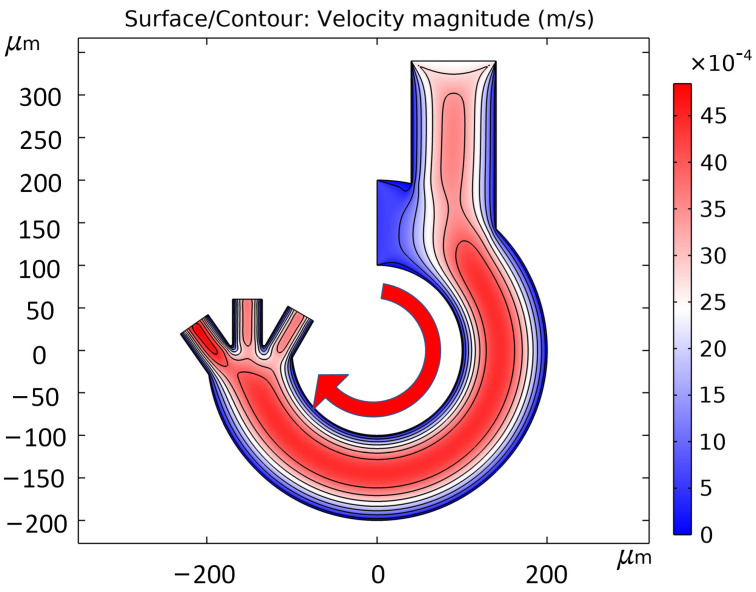
Simulation of flow rate in the channel, arrows indicate the direction of liquid flow.

**Figure 6 micromachines-14-01561-f006:**
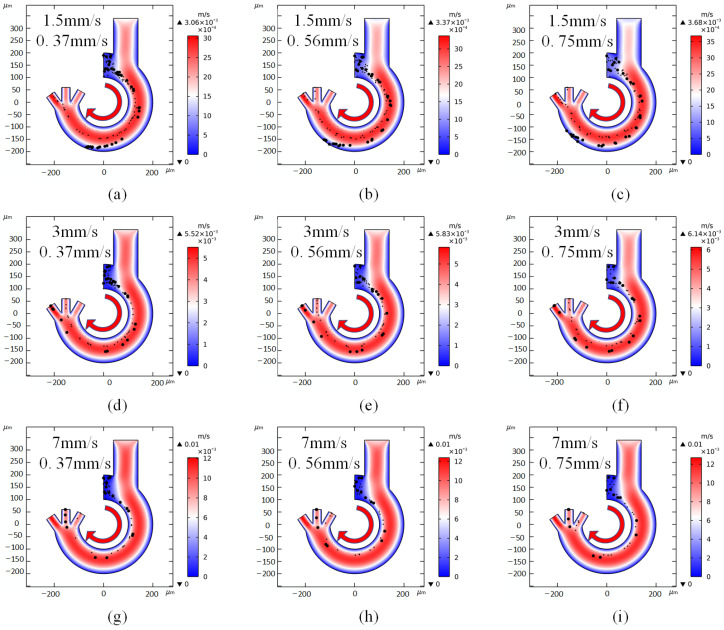
Cell sorting with different inlet flow rates. (**a**) 1.5 mm/s, 0.37 mm/s; (**b**) 1.5 mm/s, 0.56 mm/s; (**c**) 1.5 mm/s, 0.75 mm/s; (**d**) 3 mm/s, 0.37 mm/s; (**e**) 3 mm/s, 0.56 mm/s; (**f**) 3 mm/s, 0.75 mm/s; (**g**) 7 mm/s, 0.37 mm/s; (**h**) 7 mm/s, 0.56 mm/s; (**i**) 7 mm/s, 0.75 mm/s. Arrows indicate the direction of liquid flow.

**Figure 7 micromachines-14-01561-f007:**
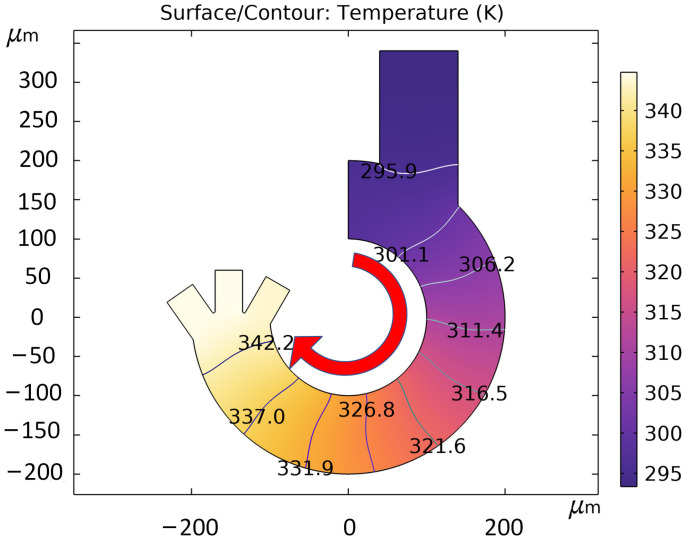
Microfluidic channel temperature simulation, arrows indicate the direction of liquid flow.

**Figure 8 micromachines-14-01561-f008:**
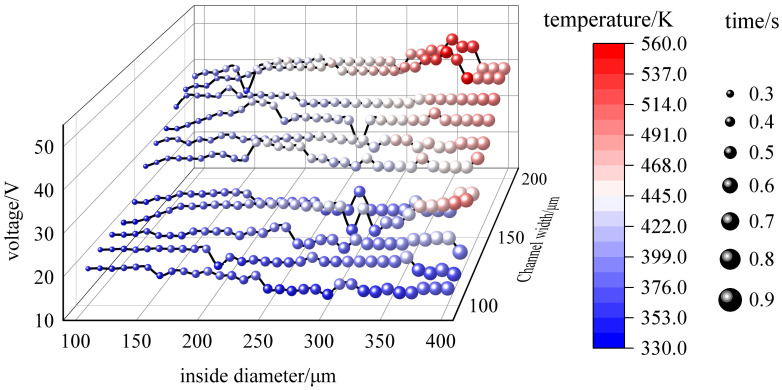
Configuration of parameters in cell sorting experiments with different microfluidic chip sizes.

**Figure 9 micromachines-14-01561-f009:**
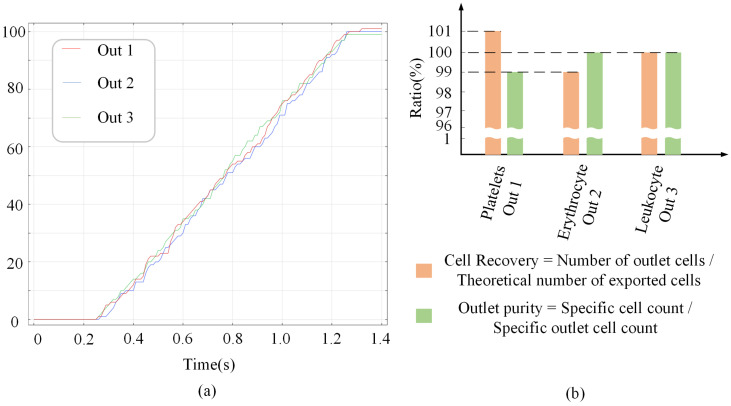
(**a**) Cell count at the outlet; (**b**) Cell recovery and purity statistics.

**Table 1 micromachines-14-01561-t001:** Microfluidic chip channel size information.

Name	Value
L1	100 μm
L2	30 μm
L3	40 μm
L4	35 μm
L5	35 μm
L6	60 μm
L7	200 μm
R1	200 μm
R2	100 μm
θ1	90°
θ2	θ2 + θ3 = 90°
θ3	θ3 = θ4
θ4	θ4 = θ3

**Table 2 micromachines-14-01561-t002:** Correlation properties of multiple cell dielectrophoresis forces.

Cell Type	*r*_2_ [μm]	*σ*_1_ [S/m]	*ε* _1_	*σ*_2_ [S/m]	*ε* _2_	*r*_2_ − *r*_1_ [nm]	Ref
Platelets	1.8	0.25	50	1.00 × 10^−6^	6	8	[32]
Erythrocyte	5	0.31	59	1.00 × 10^−6^	4.44	9	[32]
Leukocyte	12	0.65	60	2.74 × 10^−5^	6	7	[33]
A549	17	0.78	52	2.50 × 10^−5^	11.75	15	[33]
MCF-7	17	0.8	50	1.00 × 10^−6^	11	7	[27]
MCF-10A	16	0.6	100	1.00 × 10^−6^	11	7	[27]
T-lymphocyte	3.4	0.65	60	2.74 × 10^−5^	11.1	7	[28]
HT-29	11	0.72	120	3.40 × 10^−5^	11.1	4	[28]
MDA-MB-231	12.4	0.62	52	1.00 × 10^−6^	11.75	4	[28]

**Table 3 micromachines-14-01561-t003:** Simulation parameters configuration and results.

Number of Cell Types	Voltage	*σ_m_*	Inlet 1 Flow Rate	Out 1	Out 2	Out 3
4	12 V	0.055 S/m	2.4 mm/s	PLT, RBC	RBC, WBC	A549
2	12 V	0.055 S/m	60 mm/s		MCF-10A, MCF-7	MCF-10A, MCF-7
3	22 V	0.1 S/m	2.4 mm/s	PLT, TC		HT-29
3	12 V	0.1 S/m	2.4 mm/s	PLT	HT-29, MDA-231	HT-29, MDA-231

## Data Availability

The data that support the findings of this study are available on request from the corresponding author, (X.N.), upon reasonable request.

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
