# Peer review of "Design of a Low-Frequency Dielectrophoresis-Based Arc Microfluidic Chip for Multigroup Cell Sorting"

_micromachines, 2023, doi:10.3390/mi14081561_

Round 1

Reviewer 1 Report

This article presents a novel microfluidic chip based on dielectrophoresis for efficient cell sorting. Through feasibility and versatility verification using numerical simulations, the chip demonstrated successful sorting of multiple cell types with high recovery rates and outlet purity. The chip's design optimized cell separation rate, efficiency, and cell survival by minimizing Joule heat effects and reducing collisions with the channel wall. The tighter structure and integration potential with other devices make it a promising tool for biomedical research, offering a combined optimization of precision, speed, and quantity in cell sorting application. However, some revisions are recommended as follows:

(1)             The article would benefit from a more comprehensive literature review to provide a broader context for the research. Including references to recent advancements, alternative approaches, or similar studies in the field would strengthen the discussion and demonstrate the novelty of the proposed microfluidic chip. You can consider reviewing how this group of authors work on structural optimization design Frontiers in Chemistry, 2021, 9, 688442. DOI: http://dx.doi.org/10.3389/fchem.2021.688442 and work on optically induced dielectrophoresis Micromachines, 2021, 12(7), 744. DOI: http://dx.doi.org/10.3390/mi12070744

(2)          Can the authors clarify lines 191-194? They claim that when Re[KCM]>0 the negative di-electrophoretic force drives the particle to the position with the minimum electric field strength.

(3)            Line 271-273 says there are fewer methods regarding the processing and fabrication of microfluidic chips containing metal electrodes. What are they & why the suggested process being optimal for this design?

(4)          In 3.1, the author describes the chip structure design process. Here, the author mentioned that for this experiment, author used metal electrodes. But didn’t mention what type of the material of metal electrodes the author had used.

(5)          From line 304-308 the article could provide more detailed information on the selection criteria for the specific cells used in the simulation experiments. Explaining the rationale behind choosing specific cell types and their relevance to the intended applications would provide a stronger justification for the experimental design.

(6)          The article mainly focuses on sorting homogeneous cell populations, such as platelets, erythrocytes, and leukocytes. However, it would be insightful to investigate the chip's performance with more complex, heterogeneous cell populations or rare cell types. Evaluating the chip's efficiency and selectivity in sorting diverse cell populations would validate its potential for clinical applications.

(7)          In section 4.1 according to Figure 9.b & line 474-475 the cell purity at the outlet of the microfluidic chip was 99% for outlet 1, while other outlets have 100%. Can you consider discussing the reason of this impurity?

(8)          In section 4.2 line 497-500 you have added a drawback of the design. Can you propose future directions to address this limitation? For example, suggest exploring advanced techniques such as machine learning algorithms or image analysis to predict the dielectric properties of unknown cells based on certain cellular characteristics. Alternatively, discuss the potential for integrating the microfluidic chip with other analytical tools or platforms to obtain real-time dielectric property measurements during the sorting process.

(9)          The article would also benefit from addressing a potential limitation of the research methodology. For example, the study relies solely on simulation experiments conducted using COMSOL Multiphysics software (Lines 315-318). While simulations provide valuable insights, it is essential to validate the findings through experimental verification using real biological samples. This limitation should be acknowledged to ensure the robustness and reliability of the proposed microfluidic chip.

Reviewer 2 Report

This paper focus on the design of a microfluidic chip for continuous cell sorting using dielectrophoresis technology. It highlights the limitations of current microfluidic chips based on dielectrophoresis and presents the authors' objective of achieving excellent performance and high versatility. The specific approach involves a curved channel and curved double-side electrodes in the chip's structural design. The authors use software tools like MyDEP and COMSOL Multiphysics for CM factor calculations, blood cell sorting simulations, and the assessment of joule heat effects. Overall, this research addresses a significant and relevant topic in the field of microfluidics and cell sorting. The simulation layout is well structured. The primary weak point in the literature lies in its language use, leading to a noticeable decrease in readability. I would recommend publishing this paper after a thorough revision of the language.

1.    The language use in this paper can be significantly enhanced by the authors if they address the following aspects, aiming to improve the overall readability of the paper.

a.     Simplify sentence structure: The authors' use of long sentences to convey information could lead to confusion (eg: page 5 line 218-226; page 9 line 327-332; line 340-344) Break down complex sentences into shorter ones, making the ideas easier to follow. Aim for clarity and avoid overly convoluted phrasing.

b.    Create cohesion: The language use could be improved by ensuring smooth transitions between sentences and paragraphs, leading the reader from one point to the next logically and coherently.

c.     Enhancing sentence variability: I would recommend the authors introduce more sentence variety to alleviate the monotonous language use present in this paper. The paper exhibits an excessive application of similar sentence structures. (Eg: Line 77 and line 79, line 97-99 and line 99-100)

2.    The authors briefly mentioned the potential fabrication methodologies for creating the microfluidic design, either through 3D printing or traditional pattern-transferring methods using PDMS. However, the lack of detailed information in the paper raises concerns about the feasibility of the proposed fabrication process. To strengthen the paper, it is crucial for the authors to provide specific and comprehensive details to justify the viability of their chosen fabrication approach. One effective way to address this limitation is by providing direct citations of previously successful fabrication studies that have utilized similar designs or techniques. Referencing well-established research will lend credibility to the proposed methodology and demonstrate that it has been validated in prior works. Alternatively, the authors can include a comprehensive description of the specific steps involved in the fabrication process. For instance, if metal sputtering is a crucial step, providing detailed information about the equipment used, deposition parameters, and any post-processing steps will add clarity and help readers understand the procedure better.

3.    The redundant information presented in Figure 1 A and Figure 1B should be addressed to avoid duplication and streamline the content. As a suggestion, the authors could merge two figures or modify them to convey unique information. Alternatively, if both figures are essential for the paper, a clearer distinction should be made between them to justify their inclusion.

4.    In Figure 1B, the electrodes appear to be embedded within the microchannel, which raises questions about the feasibility of fabricating such a design using surface sputtering. The 3D figures provided can be confusing in this context, making it challenging to visualize the actual fabrication process.

5.    Regarding Table 1, if its content is not crucial for the main findings or discussion, moving it to supplementary information is a reasonable approach to reduce clutter in the main paper. This way, readers can access the details in the supplementary materials if needed without it interfering with the main narrative. If the information in Table 1 is essential for readers' understanding of the design, the authors could consider summarizing the key points in a paragraph within the main paper. This would ensure that the critical aspects of the design are conveyed to the readers without the need for them to refer to the supplementary materials.

6.    The parameters used in the microfluidic design, such as  numbers 88, 58, 30, and 35 have sparked my interest. However, the paper lacks sufficient information regarding the importance of these specific values and how the authors determined them.

7.    In Figures 4 to 6, I would recommend adding arrows to clearly indicate the flow direction.

8.    In the conclusion section, the authors assert that "the side electrodes used in the channel can increase the throughput of the microfluidic chip by increasing the channel depth to meet the combined optimization in precision, speed, and quantity." However, it is crucial to acknowledge potential limitations and provide a more nuanced analysis to support this claim. To strengthen this statement, the authors should clarify any restrictions or constraints associated with using side electrodes to increase channel depth. Discussing the challenges, trade-offs, or potential drawbacks of this approach will provide a more balanced perspective on its feasibility and impact.

The language use in this paper can be significantly enhanced by the authors if they address the following aspects, aiming to improve the overall readability of the paper.

a.     Simplify sentence structure: The authors' use of long sentences to convey information could lead to confusion (eg: page 5 line 218-226; page 9 line 327-332; line 340-344) Break down complex sentences into shorter ones, making the ideas easier to follow. Aim for clarity and avoid overly convoluted phrasing.

b.    Create cohesion: The language use could be improved by ensuring smooth transitions between sentences and paragraphs, leading the reader from one point to the next logically and coherently.

c.     Enhancing sentence variability: I would recommend the authors introduce more sentence variety to alleviate the monotonous language use present in this paper. The paper exhibits an excessive application of similar sentence structures. (Eg: Line 77 and line 79, line 97-99 and line 99-100)

Round 2

Reviewer 1 Report

no comments